# How Highway Landscape Visual Qualities Are Being Studied: A Systematic Literature Review

Hangyu Gao [1,*], Shamsul Abu Bakar [1,*], Suhardi Maulan [1], Mohd Johari Mohd Yusof [1], Riyadh Mundher [1] and Benxue Chen [2]

1   Department of Landscape Architecture, Faculty of Design and Architecture, Universiti Putra Malaysia, Serdang 43400, Malaysia; suhardi@upm.edu.my (S.M.); m_johari@upm.edu.my (M.J.M.Y.); gs54918@student.upm.edu.my (R.M.)

2   Department of Environmental Design, Faculty of Design, Zhoukou Normal University, Zhoukou 466001, China; cbx@zknu.edu.cn

*   Correspondence: gs58413@student.upm.edu.my (H.G.); shamsul_ab@upm.edu.my (S.A.B.)

**Abstract:** Highways play a vital role in the road transport system, connecting regions and cities in many parts of the world. It may sometimes offer scenic views or a visually appealing environment based on the availability of unique compositions of natural and man-made elements within the highway vicinity. The highway's landscapes could significantly impact the journey experience; thus, it is essential to emphasize the need to preserve a visually appealing, safe, and enjoyable highway environment. Although many studies have been conducted regarding the highway visual environment, currently, there is a lack of comprehensive understanding of perception variables that could affect viewers' preference for highway landscapes. Therefore, this study aims to understand the background of the highway landscape and identify the perception variables and their effect on the preference for highway landscapes. This study conducted a systematic review by searching for keywords in three databases: Web of Science, Scopus, and Google Scholar. The review included 37 research articles published between 1993 and 2023 that met the criteria. An additional nine relevant papers were included through a 'snowballing' approach to supplement the research and results. The results of the study focused on multiple perspectives of highway landscape views, viewers' perspectives and the diversity of highway landscape purposes, viewers' preferences for highway landscapes, the approach to preferences, and related key variables. This background knowledge deepens the understanding of visual preferences for highway landscapes and helps refine the selection of perceptual variables, establishing an essential reference criterion for professionals.

**Keywords:** visual preference; highway landscape; highway landscape preference; highway landscape preference assessment; perception variable

## 1. Introduction

Highways serve as a crucial component of the road transport system, connecting various cities, towns, and regions with networks of linkages [1]. While primarily serving the purpose of transportation, highways may also offer viewers the chance to witness impressive scenery, which adds to the enjoyment of the journey [2]. Thus, the highway landscape refers to the road's visual environment, integrating natural and cultural landscapes to create a comprehensive visual journey [3]. In other words, the highway landscape comprises diverse landscapes that viewers (including users of the highway and others who look at it in different ways) may encounter along the highway. These unique landscapes, typically characterized by the natural landscape of undulating terrain, rich flora, fauna, and water resources, or the cultural landscape of man-made elements [4], enhance the driving experience and provide opportunities for cultural appreciation and natural awareness.

In addition, highway landscapes are not just physical components but key contributors to the overall highway landscape aesthetics, offering valuable insights for aesthetic

analysis [5]. Highway landscapes also highlight that local culture should be preserved and evolving, adding intrinsic character and diversity to regional cultural identity [3,6]. This cultural aspect is pivotal in influencing visual preference and affects how viewers perceive and appreciate the scenery along their journey [2,7]. Therefore, these insights are invaluable to experts responsible for maintaining the visual integrity of highway designs and the surrounding landscapes.

### 1.1. Highway Development and Landscape Impact

Urbanization has dramatically transformed traditional rural landscapes into urban settings in recent years, fundamentally altering land use and arrangement [8]. This transformation is reflected in the changing landscape observed by highway viewers, with structures intruding into otherwise rural areas with increasing frequency [9]. As urban areas expand, the need for mass transit, including constructing and expanding roads and highways, is undeniably essential [1]. However, their construction often negatively impacts the natural surroundings, especially regarding land coverage and ecosystem improvement [10]. This trend leads to urban sprawl's cluttered, unattractive, and monotonous landscapes. Unfortunately, many highways have been constructed without adequate consideration of visual preference or quality, leading to the degradation of valuable visual landscapes [1]. Thus, highways fulfill transportation purposes and serve as markers of landscape quality changes, directly influencing viewers' perceptions and experiences of these transformations.

Meanwhile, the highway, as a medium for the combination of the natural landscape and man-made structures, emphasizes the importance of landscape design and environmental regeneration [11]. It also reflects that the combination of natural and man-made elements of the highway symbolizes the harmony between highway engineering and environmental management. However, highway development frequently undermines valuable natural and historical landscapes, leading to the loss of precious areas [12]. The construction of a major highway can profoundly alter a region's landscape ecology and scenic beauty [6]. The highway infrastructure has contributed significantly to environmental change, manifesting in alterations in land use, the loss of green areas, and changing views of and from roads [12]. Additionally, the aesthetic characteristics of minor infrastructures and vegetation alongside highways critically influence the perceived landscape quality of the roadway [6]. Hence, assessing visual preference becomes crucial when considering both the impact on the views from the highway and the aesthetic implications of the highway itself on the surrounding landscape. This consideration underscores the complex relationship between visual landscape quality and road developments [13].

### 1.2. Visual Preference on Highway Landscapes

The concept of "visual preference" can be interpreted as a psychological assessment of the observed human interaction with the environment [5,14]. This paradigm posits that individuals evaluate and react to their surroundings through emotional responses [15]. In this context of high-speed movement, the experience of driving at high speeds narrows the viewer's field of vision, primarily focusing on the immediate foreground landscape, tending to fade away, while attention is more consistently directed toward distant views [8]. In other words, highway landscapes are linear and impose limitations on viewers' appreciation in terms of visual perspective, distance, and landscape identification. Therefore, this focus shifts towards the dynamic interaction between road viewers and the scenery outside their windows while traveling at high speeds [7]. A journey becomes exhilarating for viewers when the highway presents highly preferred scenic vistas, incorporating unique landscape elements [3]. This encompasses the impact of roadside landscaping on road viewers' experiences, including their ability to navigate, control, and enjoy their journeys [5]. Hence, it is imperative for highway landscapes to provide road viewers with, or allow them to have, a comfortable, pleasant, safe, and visually appealing environment for their relaxation.

Yet, in recent years, people's preferences and perceptions of landscapes have been impacted significantly due to rapid urbanization [16]. The integration of the highway

with its adjacent landscape has significantly altered how the quality of the landscape is perceived [6]. Despite the extensive discussions on visual preferences in highway landscapes, a gap remains in the presence of unified and comprehensive perception variables for evaluating these preferences, especially in the context of the environmental impacts of urbanization and highway development. In order to address this gap, this study was dedicated to a systematic review of the existing literature on visual preference assessment in highway landscapes. Our aim is to synthesize information and identify the perception variables for evaluating highway landscape preferences. The purpose of identifying these perception variables is to deepen the understanding and perception of highway landscapes' visual preferences, aid in preserving cultural and natural values, and meet the needs of a changing society in the context of urban sprawl.

## 2. Materials and Methods

### 2.1. Keyword Selection

The keyword selection for this systematic review could be divided into four main themes: preference, visual, highway, and landscape. Keywords such as "preference" and "perception" have been included within the domain of preference. Landscape preferences are a combination of the environment's biophysical characteristics and human perceptions [17]. Perception plays a key role by providing sensory input and an initial interpretation of the environment [18], while preferences represent an individual's preferences and choices based on that interpretation [19]. In the field of landscape studies, research on visual perception investigates the fundamental concept of beauty, exploring aspects such as goodness, attractiveness, and preference. The interplay between perception and preference is complex, as each influences and shapes the other in a constant feedback loop. Hence, the review also added "perception" as a key term.

The terms "visual", "scenic", and "aesthetic" cover a broad range of visual experiences and attributes linked to personal preferences and perceptions [20]. "Visual" refers to any observable element, while "scenic" focuses on the beauty of natural landscapes. "Aesthetics" involves a deeper artistic and philosophical interpretation of beauty [21]. These terms were selected to facilitate the analysis of how individuals perceive and evaluate landscapes' visual aspects of landscape, including natural and man-made features. Furthermore, incorporating terms like "road", "roadway", or "street", in addition to "highway", would broaden the scope to enrich various road environments and their impact on visual preferences. This comprehensive approach is based on recognizing the linear character of the road landscape, which significantly affects the visual experience and shapes landscape composition [10,12]. Finally, the systematic review's keywords are summarized as follows: "preference" OR "perception" AND "visual" OR "scenic" OR "aesthetic" AND "highway" OR "roadway" OR "road" OR "street" AND "landscape".

### 2.2. Relevant Literature Screening

The methodology used to screen the relevant literature for this study was based on a keyword search and followed the guidelines of a systematic review (Figure 1). Initially, three databases—Web of Science, Scopus, and Google Scholar—were chosen to screen the literature preliminarily. Articles meeting the following criteria were selected for inclusion in this study: (1) publication between 2013 and 2023; (2) publication in an English scientific journal; and (3) classified as research papers, review papers, or conference papers. After screening and examining papers based on the given criteria, 35 out of 546 papers finally met the final requirements. To supplement this limited number, "snowballing" was employed by reviewing the reference lists and citations of the selected papers and adding six relevant papers. Although these additional papers were published earlier than 2013, they are highly pertinent to this review. Finally, 37 papers were ultimately selected—28 research papers, 1 book chapter, 2 review papers, and 6 conference papers published between 1993 and 2023.

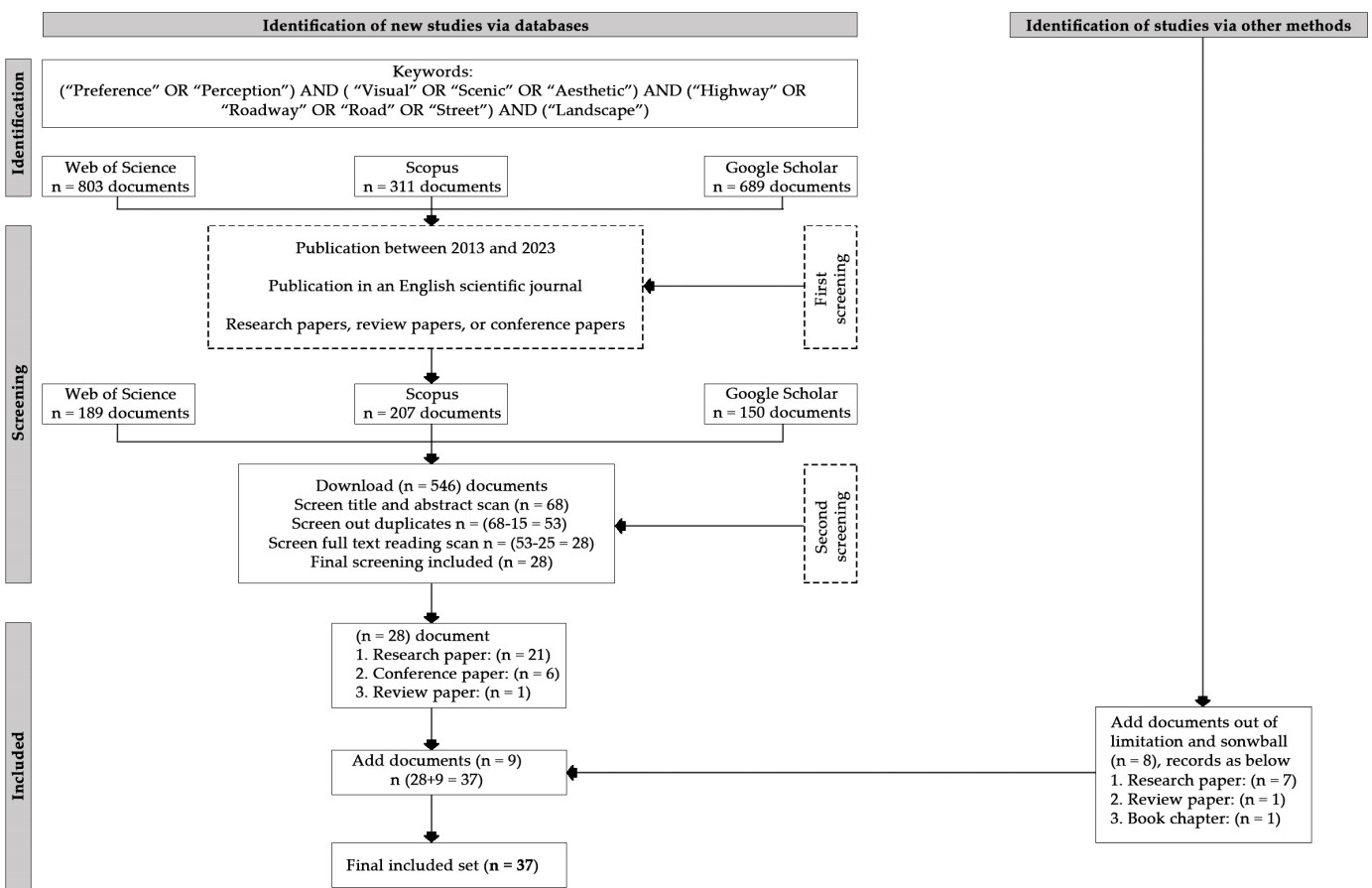

**Figure 1.** Flowchart describing the literature-screening process for systematic search reviews.

### 2.3. Data Collection

For this study, a thorough analysis and reading were conducted on the selected articles, and the related information was gathered and organized in an Excel spreadsheet. The recorded information consists of several elements, such as the types of roads, the landscape character in which the road is embedded, the types of viewers, the assessment methods, the response format, the media used to represent the view, and the criteria or variables used in the assessment. Appendix A, Table A1 presents a comprehensive overview of the collected data.

### 2.4. Data Analysis

This study presents the synthesis of the information referred to in Appendix A of Table A1, with the objective of strengthening the connections between the literature and enhancing the depth of the findings. Initially, it explored the interactions between highways and their surrounding landscapes from multiple perspectives by integrating the various types of roads with the landscapes in which they are located. Secondly, it considered the interaction between the viewer's perspective and the research purposes across different roadway contexts. Next, an in-depth understanding and detailed analysis of viewers' preferences in highway landscapes were provided. Additionally, in order to achieve a comprehensive understanding of the research methodology, the evaluation techniques, response formats, and the media used to present the views were examined. The discussion of evaluation criteria or variables was intended to identify unified and comprehensive perception variables for assessing highway landscapes.

## 3. Results

In order to better understand the assessment of perceptual variables on highway landscapes, it is essential to have a clear knowledge of the contextual aspects of highway landscapes (discussed in Section 2.4). The results of the study are, therefore, divided into the following categories, which are discussed in the following sections:

- Intersecting landscapes and a multidimensional exploration of highway landscapes;
- Understanding perception and preference in the visual dynamic of roadside landscapes;
- Understanding a viewer's preference in highway and roadside landscapes;
- Assessing highway landscape visual preferences through various approaches;
- Identifying key perception variables in assessing the visual preference of highway landscapes.

### 3.1. Intersecting Landscapes and Multidimensional Exploration of Highway Landscapes

Highway landscapes are the main focus of this study; however, adding different types of roadway landscapes broadens the area and enhances meaningfulness with respect to understanding more in-depth views of highway landscapes. The highway exhibits diverse landscapes, including natural landscapes [2–5,8,10,12,22–26], such as mountains, forests, water features, and roadside vegetation. Meanwhile, cultural landscape elements within these highway landscapes serve an integral purpose. For instance, the historic structures within the view angle of a highway enhance the architectural characteristics of the highway and maintain a relationship with viewers, linking them to the heritage of a region [2,25]. Furthermore, the topographical variations further add visual interest as highways traverse varied landscapes, from mountainous landscapes with high peaks and ridges, to basins, enhancing the highway travel experience [4,25]. At the same time, such scenes are also depicted on scenic routes, as explored in studies [27,28], which highlight valleys, vegetation, water features, stone walls, pastures, historic residences, a blend of natural and pastoral scenes, forested areas, and open vistas. This demonstrates the richness and diversity of what the highway can do as a tourist roadway.

When highway landscapes pass through rural or urban areas, they provide another multifaceted perspective. For example, the highways traversing rural areas reveal landscapes dominated by agricultural lands and small towns [3,5,10,13,29], offering a glimpse into the serene rural life seen on rural roads [30,31]. The experience is different, however, as the leisurely pace of rural roads allows for a more relaxed appreciation of the scenery. In contrast, urban highways pass through transition zones that are distinct from the bustling commercial, residential, and green spaces of urban streets [32–36].

Studies [5,9] demonstrate how highways can combine these two landscapes, mirroring the rural–urban fringes discussed in [37], displaying characteristics of an integrated landscape. Notably, certain regions along highways display distinct landscapes, such as palm oil estates [3] and volcanic landscapes [12], adding depth to the visual narrative for viewers. Therefore, highway landscapes represent a dynamic interplay of natural beauty, historical richness, and the built environment. The variety of landscapes emphasizes the highway's role as a connecting link between destinations, different environments, and cultural experiences.

### 3.2. Understanding Perception and Preference in the Visual Dynamic of Roadside Landscapes

The visual quality of human surroundings, especially within highways, urban streets, and rural roads, profoundly impacts daily experiences, comfort, and satisfaction. To this end, numerous studies have explored the perceptions and preferences of people affected by these landscapes—such as drivers, passengers, and pedestrians, as illustrated in Table 1. For instance, studies focusing on highway landscapes have primarily assessed the visual quality experience and preferences of the public [5,8,9,24] or highway users [3], emphasizing the recognition of the importance of their perceptions. Similarly, studies involving pedestrians [32] or residents [36] have further aimed to evaluate the aesthetic aspects of specific streets or roads in urban settings. Notably, a study on urban roads

has investigated how the presence of poplar trees enhances the aesthetics of urban road landscapes, as perceived by 35 university faculty members [33].

In rural settings, studies [30,31] have evaluated the visual quality of rural roadside landscapes and focused on understanding public or specific group preferences for various rural landscapes. For mixed urban and rural areas, as seen in [37], the aim is to explore residents' preferences for agricultural and development patterns along urban and rural roads through residents, with the simultaneous intention of promoting economic development and improving the visual quality of the roadside. Scenic roads have also been analyzed for how different landscape components contribute to visual quality, comparing residents' preferences for natural, cultural, and mixed landscapes [24]. Moreover, study [28] has focused on the aesthetic characteristics that contribute to the success of scenic road design, examining motorists' perceptions. Therefore, those studies emphasize the importance of understanding the perceptions and preferences of different groups of people towards these landscapes.

The exploration of landscape preference extends beyond visual quality, incorporating the interplay between sensory experiences and environmental perception. Specifically, with regard to studies targeting specific demographics, such as university students aged 18 to 27 [29] and online social media users aged 18 to 47 [13], these studies have explored the interactions between traffic noise and highway visual perception, intending to help us understand how traffic noise affects people's visual perception of highways. Moreover, a comparative analysis between expert opinions and the general public's views further illuminates the significance of comprehensively adopting diverse approaches to understanding and assessing scenic beauty in highway landscapes [2,6,7,10]. However, studies [35,38] have mentioned testing new methods' effectiveness through comparison. Therefore, such a comparison provides a more profound comprehension of landscape aesthetics in highway landscapes and encourages the innovation of new approaches.

Studies lacking direct participant (only expert) involvement in highway landscapes [4,9,11,12,22,23,25,26,39] have aimed at creating a new approach for evaluating and improving landscapes around highways, emphasizing aesthetics, environment, and safety. Similarly, the urban-centered study employs innovative techniques such as deep learning [40] and other machine-learning approaches [34]. On the other hand, study [41] has summarized the visual characteristics to improve the understanding and assessment of urban landscapes' visual quality and perception.

Overall, these studies affirm that all respondents, considered as viewers of the roadside landscape, play a pivotal role in defining research directions. Perspectives and experiences from different groups, such as drivers, passengers, and specific age demographics, provide unique ideas for understanding and evaluating roadside landscapes' visual and environmental quality. These studies not only reveal the viewer's visual preferences for highways and urban streetscapes, but also highlight the need to incorporate diverse perspectives into the evaluation process.

**Table 1.** Show the differences between viewers and purposes.

| Type of Roads | Viewers | Purposes | References |
|---|---|---|---|
| Highway | Highway users or the general public | Assessing experiences and preferences regarding highway landscapes' visual quality. | [3,5,8,9,24] |
| | College students (18–27), users online (18–47) | Examining how traffic noise impacts visual perception of highways. | [13,29] |
| | Experts and the general public | Adopting diverse approaches in assessing scenic beauty. | [2,6,7,10] |
| | None | Exploring new methods for highway landscape evaluation. | [4,9,11,12,22,23,25,26,39] |

**Table 1.** *Cont.*

| Type of Roads | Viewers | Purposes | References |
|---|---|---|---|
| Urban street | Pedestrians, residents | Assessing the visual quality of specific streets or roads. | [32,36] |
| | Academic staff | Understanding how poplar plantings affect people's visual preferences and perception of landscape quality. | [33] |
| | Experts and the general public | Exploring new methods for people's perceptions of the urban road landscape. | [35] |
| | None | | [34,40,41] |
| Scenic road | Residents | Assessing the significance of natural and cultural features in landscape preferences. | [24] |
| | Motorists | Exploring how the design of the road influences motorists' perceptions and emotional responses. | [28] |
| | Experts and the general public | Exploring new methods for enhancing the driving experience. | [38] |
| Rural road | Public, motorcyclists | Assessing the visual quality of the road landscape. | [30,31] |
| Urban–rural road | Local residents | Understanding resident preferences for agriculture and development along mixed urban and rural roads. | [37] |

### 3.3. Understanding Viewer's Preference in Highway and Roadside Landscape

Regarding scenic road selections, the public's judgment is influenced by various factors, demonstrating strong personal preferences throughout the decision-making process [38]. In most cases, such strong expressions of individual preferences can lead to a consensus, revealing the public's widespread appreciation or aversion towards specific types of landscapes. In highway landscapes, viewers strongly prefer natural elements such as farmland, limestone hills, mountains, and other natural characteristics [3], as well as distinctive bodies of water and building features [2]. These results are also found in scenic roads [27], confirming an appreciation for natural aspects such as water and vegetation and highlighting the value of cultural elements in their role in preferences, such as stone walls and historic residences. Conversely, cultural landscape elements like towns, residential, and industrial areas have low visual appeal [2,3], perhaps due to inadequate environmental integration or insufficient maintenance. Such results appear consistent with rural road landscapes, landscapes with roadside settlements, and commercial structures, which are less attractive [30].

The composition of roadside vegetation also significantly impacts visual preferences. Viewers tend to prefer diverse and colorful vegetation types [5] over a uniform mix of plant species [24]. Scenic road design elements, such as reverse curves and the strategic use of color and texture in plantings, significantly contribute to the positive aesthetic experiences of viewers [28]. The preferred combination of vegetation types consists of trees in the background, followed by shrubs in the middle, and grass and flowering plants in the foreground [5,24]. This preference extends to rural roads, where landscapes with water bodies in the foreground and wooded backgrounds are particularly valued [31]. Some studies [9,33,37] have also provided additional evidence of the role of trees in enhancing the visual quality of roads, emphasizing their indispensable contribution to roadside aesthetics.

However, roadside trees and shrubs can negatively impact sightlines on major roads [42], and neglected vegetation maintenance can deteriorate visual experiences on rural roads [30]. These findings highlight the importance of balanced ecological management to maintain both aesthetic appeal and safety. Particularly, study [7] has also emphasized that the openness of the field of view is crucial in enhancing the viewer's aesthetic preference in a dynamic environment. Interestingly, there is a discrepancy between the results obtained from the static questionnaire survey and the dynamic simulation of the highway landscape. While water bodies are usually seen as enhancing preference, their appeal diminishes in

dynamic assessments, suggesting that motion could influence landscape perception [7]. Similarly, cultural elements usually considered negative, such as bridges, are perceived positively in dynamic contexts.

Furthermore, both frequency and noise levels affect visual preferences on the highway. According to study [5], infrequent viewers express higher satisfaction with the type and layout of roadside vegetation, while frequent highway users prefer a wider variety of colorful vegetation. Elevated traffic noise may affect the visual experience, especially with high traffic volumes [13]. The study indicates this auditory impact heightens the visual disturbance by an average of 11.6 on a scale of 1 to 100 across different scenarios. This effect slightly strengthens at 300 m (compared with 100 or 200 m), indicating a subtle but noticeable enhancement in the interaction between auditory and visual perceptions with increased distance from the noise source.

In summary, highway landscape preferences are influenced by natural and psychological factors, personal experiences, and practical considerations. Various factors, including diversity, continuity, management level, natural feeling, and aesthetics of roadside greening, influence the visual quality of roadside landscapes [36]. With proper management and maintenance, it is possible to create highway landscapes to meet the aesthetic needs of different viewers and enhance the overall experience of road travel.

### 3.4. Assessing Highway Landscape Visual Preferences through Various Approaches

Through a systematic review of the literature, three main approaches have emerged in exploring methods to assess highway visual preferences, incorporating urban, rural, and scenic road landscapes: the viewer's perception approach, the expert approach, and a hybrid of the two approaches (Figure 2).

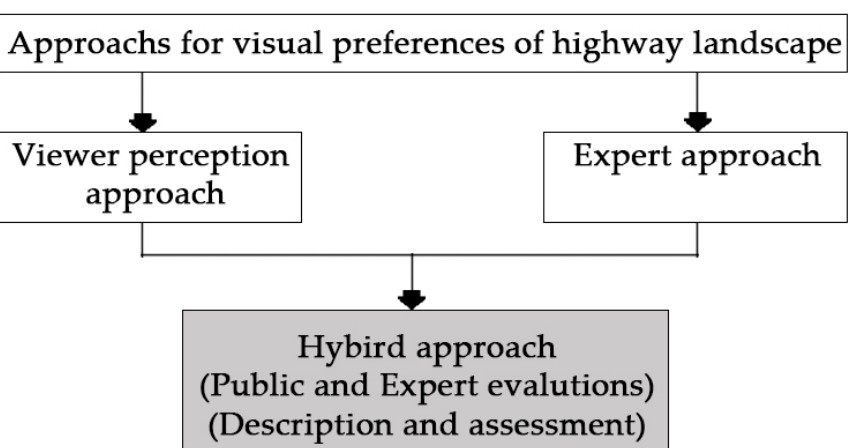

**Figure 2.** Three different approaches.

### 3.4.1. Perception Approach

Assessing highway landscapes is a complex process, significantly influenced by its subjective nature, which heavily relies on an individual's perception and response to the landscape [5,24]. Typically, individuals tend to prefer natural landscapes that are easily understandable and have various features; however, preferences for such landscapes can vary widely depending on the viewer's background [31]. This approach relies mainly on quantitative methods involving a specific group or the general public. In a viewer perception approach as applied in landscape studies, mean ratings are utilized to evaluate individual perceptions of highway landscapes. Therefore, the reliability of the results may be higher when using this approach than relying solely on experts [20].

The perception approach in assessments generally includes surveys and the evaluation of photographs selected by researchers, which participants are asked to rate. However, assessing how image quality influences viewer ratings introduces a challenge, as it may introduce bias through the researcher's selection of landscape images [8]. To address

this concern, study [8] has proposed conducting individual photographic surveys using disposable cameras, enabling participants to capture landscapes they cherish and, thus, avoid the bias that might arise from pre-selected images. To closely mimic realism further, some studies, including [13,29], have adopted computer visualization techniques and edited audio recordings to simulate highway scenarios in a laboratory setting. This approach is intended to enrich the study's context and potentially increase the interpretability of the findings related to visual and auditory perceptions of the landscape. Overall, this approach based on perception provides a quantifiable and reliable way to produce accurate results. It comprehensively explains viewers' perceptions of the scenic beauty of the highway [24] and understands the vital influence of the biophysical environment along the road. However, it faces challenges such as the longer time required for collecting surveys and handling the data.

### 3.4.2. Expert Approach

The expert approach favors an objective methodology that converts the physical characteristics of the landscape into critical indicators that influence visual quality [20]. Grounded in the experience and judgment of experts, this approach involves a systematic assessment and recommendations process on landscape characterization regarding established norms and standards [20]. Experts typically elaborate and explain features of the highway landscape, then evaluate them according to the relevant criteria [10]. This approach represents a cost-effective strategy [20] and is widely adopted [27], efficiently achieving relevant objectives. However, it does face some remaining shortcomings. This approach primarily relies on experts' opinions, and the variables used by the experts are not revealed to be good predictors of preference [2,10]. Experts are also assumed to be broadly consistent in their landscape assessments while overlooking the viewer's subjective perception [2]. Such limitations may affect the accuracy and reliability of the results. While studies [34,40] have applied deep-learning models to innovate public assessment systems and examine urban landscape perceptions, these do not inherently assure the validity of the variables used in these models.

Moreover, the expert-centric approach somehow fosters investigating and implementing new assessment approaches. For example, study [11] has incorporated expert analysis with a holistic methodology that synthesizes quantitative and qualitative methods, including set-pair analysis, to systematically evaluate the highway landscape's quality through a structured blend of various criteria and indicators. Study [25] has focused on the impacts of visibility and landscape management, using a GIS-based viewshed analysis and visual magnitude analyses to carry out a visibility analysis, providing unique information about the topography visible from the highway. Study [26] has combined expert analysis with data extraction methods using Google Maps and Google Earth for landscape elements, including evaluating the visibility and prominence of landscape features along road segments. Generally speaking, the expert-centric approach often results in the adoption of newer technologies. However, it is essential to note that these choices are still influenced by the background and experience of these experts, and the outcomes can be somewhat controversial.

### 3.4.3. Hybrid Approach

Study [20] has mentioned that this hybrid approach could be considered more informative because it facilitates a direct comparison of the two approaches (expert and viewer perception), aiming to understand and resolve their inconsistencies systematically. However, only a few studies in the reviewed literature have adopted this approach. For example, study [35] has integrated expert analysis to develop the indices and viewer perceptions to validate the assessment method. Study [6] has compared the visual highway landscape quality obtained through a systematic photo analysis with the landscape preferences determined using a survey. Similarly, study [10] has investigated how experts use descriptors to evaluate landscape quality and correlate their assessments with the ratings

of landscape beauty from untrained observers. Moreover, studies [2,42] have evaluated the quality and differences of scenic spots by combining expert analysis and viewer perception. A notably innovative application of this approach is found in study [38], where the development of the Autobahn system—a tool for generating scenic routes using Google Street View images—is evaluated against traditional route-planning algorithms. These approaches comprehensively compare the results obtained by both approaches. However, study [7] has combined a survey of user perceptions with expert analysis. It has employed experts to quantify visual attractiveness objectively through physiological measurements, which further explains the research results. Additionally, certain studies of the hybrid approach [7,42] have also used restoring real scenes to better understand visual preferences in real moving environments. Generally speaking, it allows for a nuanced evaluation of landscape quality, bridging the gap between objective expert assessments and subjective viewer experiences. This hybrid approach can yield good results but requires more time, energy, and financial resources.

### 3.5. Identifying Key Perception Variables in Assessing Visual Preference of Highway Landscapes

The landscape visual preference assessment approach has three main assessment variables and criteria: physical, aesthetic, and psychological [6]. The physical variable primarily emphasizes the significance of environmental elements and their interactions, including both natural and cultural elements [30]. The aesthetic descriptor arranges the landscape's physical features into visual components such as shapes, colors, lines, and compositions [5]. In other words, both physical and aesthetic variables are based on the physical characteristics of the landscape. Conversely, psychological variables delve into how the landscape's physical attributes impact viewers' perceptions [32,33], highlighting the importance of these physical and aesthetic features in shaping the viewer's visual perception [6].

In highway landscape assessment, these landscapes rely heavily on physical features to trigger visual and psychological evaluations, thereby playing a pivotal role in the assessment process [7]. Thus, study [2] has suggested that an initial step in highway landscape preference assessment should involve thoroughly categorizing and describing the observed landscape. This is followed by considering variables related to visual perception before proceeding with the assessment.

The initial stage in highway landscape assessment, called highway landscape characterization, involves defining, classifying, and labeling a landscape area's distinctive highway landscape character [43]. This stage requires a detailed description of each area's character, distinguishing it from others based on the diversity, organization, and spatial arrangement of landscape features [30]. This process ultimately gives each area a distinct character that sets it apart from its surroundings [2]. Reviewed studies [27,31] utilize the landscape's physical variable for visual evaluation, highlighting that physical factors can influence the viewer's assessment of landscape elements. However, study [44] has argued that using the landscape's physical attributes as the sole assessment criterion is inadequate. It should be viewed as an objective unit that cannot be used in isolation to explain visual preferences. Additionally, cultural features of the landscape can be considered secondary and potentially negative [27]. Therefore, identifying highway landscape types is a key step in understanding visual preferences [2]. As previously mentioned (discussed in Sections 3.1 and 3.3), highways traverse diverse areas, such as mountains, forests, cities, and villages, and these areas exhibit changes in landscape character and topography. The classification of highway landscapes is influenced by various factors, including land use (such as agricultural, residential, commercial, and industrial), topography (such as mountains, hills, and plains), and land cover (such as vegetation, bodies of water, and man-made structures).

The second stage assesses the features or elements of the highway landscape that are, then, incorporated into appropriate management, planning, and preservation options. The process includes considering variables related to visual perception. Experts and viewers

assess these variables to determine the visual preferences of the highway landscape. There-fore, identifying key variables relevant to visual perception becomes critical in this process.

According to study [2], human vision is vital in understanding and exploring the environment and assessing the current situation and possible future changes. The legibility and coherence of the environment facilitate comprehension, while complex and mysterious environments encourage exploration [2]. Similarly, naturalness is an important concern in viewers' understanding and exploration of the environment, with many programs pointing to natural attributes as a central element in viewers' ideals and preference choices [10]. Al-though the study's results [10] have shown that naturalness does not significantly improve the ability to predict preferences, many studies include naturalness in their evaluation framework. Moreover, visibility (openness) is highlighted as a key criterion for landscape preference, with the literature reviewed consistently emphasizing the impact of openness on preference [2,7]. Objective indicators are used to assess street visual quality intelligence, with greenery and the sky view responding to the openness and closure in the street view [34], which is enough to show the importance of openness. Finally, imageability further enriches this discussion, denoting the distinct characteristics that make an area memorable or easily recognizable [32], facilitating landscape differentiation. Study [34] has indicated a moderate correlation between imageability and visual quality, suggesting that streets with distinct imageability are perceived as having a higher visual quality.

In sum, highway landscape preference assessment primarily revolves around seven variables: naturalness, openness, complexity, coherence, legibility, mystery, and image-ability. These variables are organized into three main criteria: highway-landscape visual ambiance (openness and naturalness), highway-landscape visual composition (complexity, coherence, legibility, and mystery), and highway-landscape visual impression (imageabil-ity). Therefore, the whole process is displayed in Figure 3.

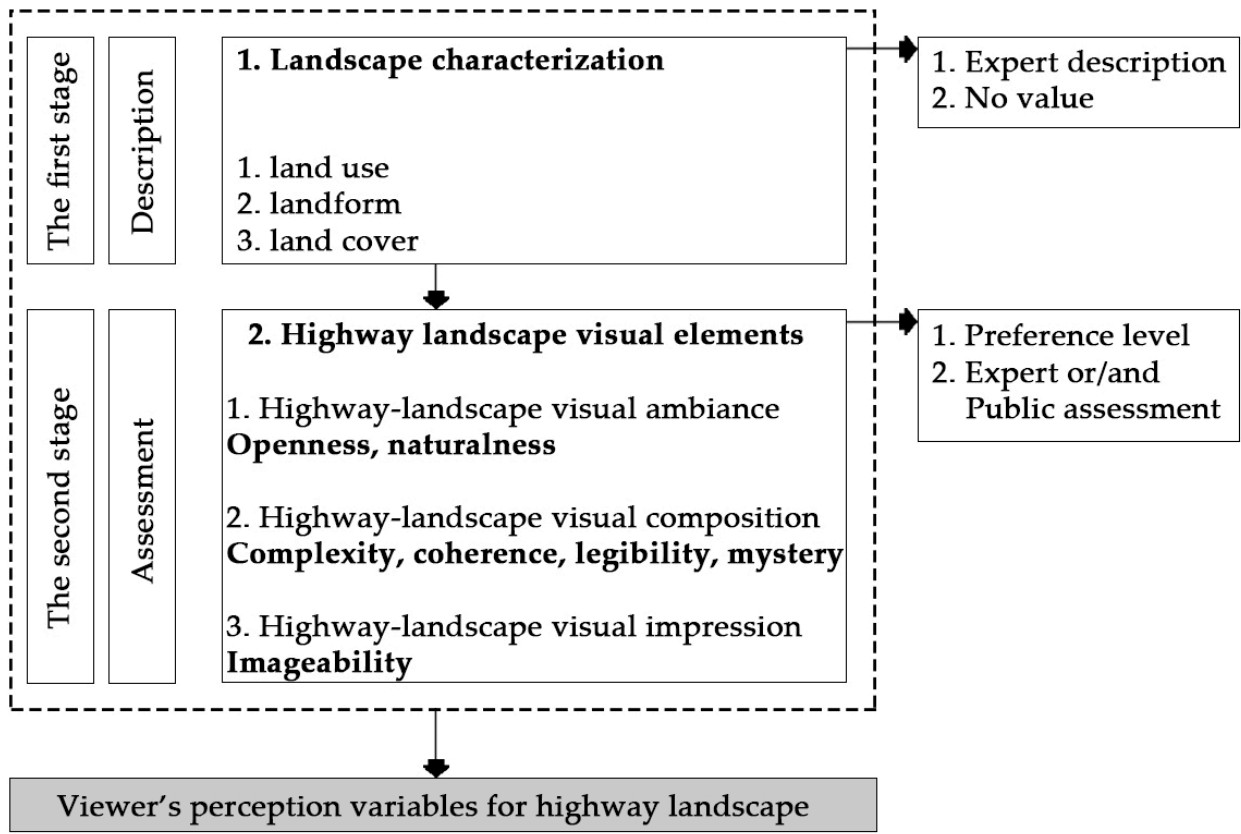

**Figure 3.** Landscape characterization and perception variables of highway landscape.

## 4. Discussion

The study initially established an understanding of the landscapes along or around highways and other roads, exploring research objectives and preferences to comprehend the characteristics and elements of highway landscape preferences. Furthermore, this study not only analyzed the methods used to identify these landscape preferences, but also provided a detailed discussion of the variables involved in the methods. In particular, a more complete and unified set of perception variables for highway landscape preferences was summarized and presented, which might have critical applications for future research on highway landscape preferences. Over the years, as the demand for grade-separated transport networks has continued to grow, there has been a significant increase in interest in the physical aspects of highways and their potential impact on viewers [5]. The aesthetics of the landscape perceived by a viewer of a highway depends on the physical and psychological distance between the viewer and the landscape [5]. As a result, transport networks play a crucial role in shaping the landscape character and facilitating human interaction with the landscape.

In this period of profound human impact on the land, roads have introduced novel perspectives on environmental interaction [6]. Landscapes along roadsides, encountered daily by individuals, play a crucial role in introducing individuals to new regions, encouraging the exploration of the surrounding environment, and shaping their perception [42]. Highways, in particular, emphasize the importance of landscape design and environmental regeneration as a medium for integrating natural landscapes and man-made structures [11]. The complexity and diversity in highway landscapes emphasize that they are about more than just transport. The landscape presents diverse natural and cultural features, including forests and historical roadside architecture, offering travelers a distinctive visual and experiential journey. The highway, thus, serves both transportation and leisure purposes.

In terms of their purpose and participants, these studies reveal the importance of considering a variety of perspectives to assess aesthetic aspects and the impact of environmental change. On the one hand, most studies with participants emphasize the subjective nature of a visual preference assessment and the importance of including a wide range of perspectives. On the other hand, studies lacking participants (experts only) highlighted the importance of innovation. However, the hybrid approach promotes a more comprehensive discussion of aesthetic values and stimulates the creation of novel approaches to landscape valuation. Beyond this, investigating visual preferences encompasses not only aesthetics, but also the interaction between sensory experience and environmental perception, particularly in the context of traffic noise.

The visual preference of the highway landscape is pivotal to viewers' experience using the highway, influenced by physical factors, such as patterns of land cover and land use, and individual psychological factors, such as mental information [2]. The assessment of landscape aesthetic preferences is highly subjective due to individual psychological factors; i.e., people's interactions, past experiences, and current landscape characteristics differ in their evaluation of the attractiveness of the same landscape [31]. In general, viewers prefer nature, natural diversity, and historical significance in highway landscapes, while being critical of poorly integrated or maintained cultural elements. This is because highway infrastructure causes anthropogenic changes in the surrounding environment, leading to land use and vegetation alterations and a reduction in greenery [12]. Neglect in managing these landscapes can lead to a cluttered and disorganized appearance, negatively affecting viewers' preferences. Furthermore, the preference for motorway landscapes is influenced by various factors, including the level of highway use, external dynamics such as noise, and the complexity of the factors influencing preference.

The growing interest in the visual quality of highway landscapes began primarily in the mid-20th century in America [10]. During this period, many tourist attractions in America were closely associated with automobile travel, making the highway landscape pivotal to drivers and passengers. As leisure activities on the road increased, the evaluation of the quality of the highway landscape between destinations became increasingly

important. It was during this era that the visual assessment of highway landscapes began to be recognized. Three different methods were used consecutively to assess highway preferences. The viewer perception approach is based on subjective experience and aims to reach a consensus on landscape quality. The expert approach uses established criteria and expert knowledge to analyze natural landscape features objectively. The hybrid approach combines objective analyses with subjective experiences to comprehensively understand. This approach emphasizes the multifaceted nature of landscape aesthetics and the importance of using different strategies for assessment.

A highway landscape visual preference assessment comprises a systematic and scientific inquiry into the current state of the landscape environment. The evaluation considers formal aesthetics and highway viewer's values, thus offering a comprehensive overview of the landscape environment along the roadside. Assessing landscape vision is inherently complicated due to the task of capturing the nuanced human perceptual experience, specifically while dealing with transient landscape elements [10]. Human visual cognitive behavior involves more than just gathering external information; it is a complex mental process combining judgments and noticing visible specifics. Scientific evidence emphasizes the significance of the roadside landscape and its visual attributes for road viewers' perceptions [33]. A valid highway landscape assessment system offers a comprehensive assessment and understanding of the environmental conditions in and around the highway [11]. Therefore, this study attempts to propose comprehensive variables for characterizing and evaluating highway landscape characters and their preferences.

## 5. Limitation and Future Studies

This study systematically reviews viewers' preferences toward the highway landscape. However, this study still has some flaws. Firstly, the study only used qualitative analysis and lacked quantitative analysis. Therefore, it is recommended that future research includes quantitative analysis, such as meta-analysis, further to investigate the published topic in the related field. Next, the number of keywords was limited. In future research, keywords such as "preference criteria", "preference factors", and "preference impacts" could be included to enhance the number of articles and the understanding of preference variables. Lastly, while the study identified the variables influencing highway landscape preferences, it lacked an in-depth discussion, assessment, and validation of each variable. Therefore, it is crucial for future research to explore these variables more deeply, offering adequate theoretical and practical definitions for the preference assessment.

## 6. Conclusions

This study analyzed 37 papers on landscape preferences for highways and other roads, uncovering that the perception variables of openness and naturalness for the visual ambiance; complexity, coherence, legibility, and mystery for the visual composition; and imageability for the visual impression of the highway landscape have an effect on visual preferences for highways. Viewers' visual preferences may be intuitive and influenced by their personal favorites. Our findings suggest that, while natural landscapes are generally preferred, it is essential to recognize the significance of cultural landscapes for a comprehensive understanding of highway landscape preferences. Although the study has improved the understanding of highway landscape preferences through more refined perceptual variables, a crucial next step involves validating and critically assessing these variables. Employing a preference/scenic quality research approach, professionals and viewers can simultaneously rate the same highway landscapes. This allows the variables and relationships between the differences to be explored and validated. In conclusion, the findings of this study refine the completeness of the perceptual variables of highway landscapes and establish the relevant reference criteria for professionals.

**Author Contributions:** Conceptualization, H.G., S.A.B. and S.M.; methodology, H.G. and M.J.M.Y.; data analysis, H.G., S.A.B. and S.M.; writing—original draft preparation, H.G. and R.M.; writing—review and editing, M.J.M.Y., S.M. and B.C.; visualization, S.M. and R.M.; supervision, M.J.M.Y. and B.C. All authors have read and agreed to the published version of the manuscript.

**Funding:** This research received no external funding. The APC was funded by the Scenic American.

**Data Availability Statement:** No new data were created or analyzed in this study. Data sharing is not applicable to this article.

**Conflicts of Interest:** The authors declare no conflicts of interest.

## Appendix A

**Table A1.** Summary table of all 37 documents.

| Ref. No. (No) | Type of Road | Landscape Character in Which the Road Is Embedded | Viewers | Views | Purpose of the Study | Assessment Method | Response Format | Media Used to Represent the View | Criteria or Variables |
|---|---|---|---|---|---|---|---|---|---|
| [1] (1) | Highway | Not detail the specific surrounding landscape character | NO | F | Identifying landscape characters | ER | - | - | None |
| [2] (2) | Highway | Rounded glacially sculpted hills and ridges, second-growth forests, and so on | N | T | Assessing scenic quality | PR and ER | PQ | CP | Coherence, complexity, legibility, mystery, openness, smoothness, and locomotion |
| [3] (3) | Highway | A variety of natural and cultural elements | U | F | Evaluating users' preferences | PR | PQ | CP | Natural landscape elements and cultural landscape elements |
| [4] (4) | Highway | Complex mountainous terrain, hydrology, and geology | NO | F | Assessing landscape character | ER | - | - | Function, aesthetics, ecology, and safety |
| [5] (5) | Highway | Urban residential areas and countryside agricultural land | U | F | Assessing scenic beauty | PR | S | CP | Variety of vegetation, colorfulness, vegetation type and combination, and perceived importance of various elements |
| [6] (6) | Highway | Not detail the specific surrounding landscape character | N | T | Assessing landscape character | PR and ER | S | CP and PM | Physical, aesthetic, and psychological attributes |
| [7] (7) | Highway | Not provide detailed descriptions of each landscape character | NO | F | Assessing scenic quality | PR and ER | OS | CTA | Visual attraction factors and physiological signals |
| [8] (8) | Highway | Two large physiographic units | U | F | Assessing visual impact | PR | S | CP | Various elements of the highway landscape |
| [9] (9) | Highway | Not detail the specific surrounding landscape character | NO | F | Assessing visual landscape | PR | ONI | CP | Color tendencies, materials, and recognition degrees |

**Table A1.** *Cont.*

| Ref. No. (No) | Type of Road | Landscape Character in Which the Road Is Embedded | Viewers | Views | Purpose of the Study | Assessment Method | Response Format | Media Used to Represent the View | Criteria or Variables |
|---|---|---|---|---|---|---|---|---|---|
| [10] (10) | Highway | Diverse landscapes, including mountain-valley conditions, and so on | N | T | Assessing scenic beauty | PR and ER | SL | CP | Naturalness, vividness, variety, and unity |
| [11] (11) | Highway | None | NO | F | Assessing landscape character | ER | - | PM | Indicator system that includes landscape features, environmental harmony, and so on |
| [12] (12) | Highway | Natural and introduced flora/fauna, colors, lines, patterns, and human interventions | NO | F | Analyzing landscape preferences | ER | ONI | OB | Criteria such as the presence of man-made obstacles, perceptual units, and so on |
| [13] (13) | Highway | Natural and semi-rural residential landscapes | U | F | Assessing visual impact | PR | S | CTA | Various traffic conditions, two types of landscapes, three viewing distances, and sound condition |
| [20] (14) | - | - | - | | Reviewing in assessing visual landscape quality | ER | - | - | - |
| [22] (15) | Highway | A diverse landscape, including vast forests and mountain ranges | NO | F | Assessing landscape character | ER | ONI | CP | Landscape visual qualities and natural features |
| [23] (16) | Highway | Relief and vegetation type | NO | T | Assessing landscape character | ER (CM) | - | GIS and PM | Visual landscape character, such as coherence, complexity, and so on |
| [24] (17) | Highway | Not detail the specific surrounding landscape character | U | F | Understanding scenic quality | Pr | PQ | CP | Types of vegetation on road verges |
| [25] (18) | Highway | Mountains, lakeside cliffs, dense forests, diverse topographical features, and several rivers | NO | T | Assessing landscape character | ER (CM) | - | CTA | Visual magnitude |
| [26] (19) | Great ocean road and highway | Diverse elements such as individual trees, forests, water, beach shores, etc. | NO | F | Assessing landscape character | ER (CM) | - | CP | Road segments, including road orientation, relative elevation, openness, distance to horizon, and the presence of specific elements |

**Table A1.** *Cont.*

| Ref. No. (No) | Type of Road | Landscape Character in Which the Road Is Embedded | Viewers | Views | Purpose of the Study | Assessment Method | Response Format | Media Used to Represent the View | Criteria or Variables |
|---|---|---|---|---|---|---|---|---|---|
| [27] (20) | Scenic road | Hilly terrain, three sizable rivers, valleys, and extensive wetlands | N | F | Assessing landscape preferences | PR | PQ | SL | Water, open views, mature trees, stone walls, geologic features, agricultural uses, historic areas, and residences |
| [28] (21) | Scenic road | A mix of natural and pastoral scenes, forested areas, and open vistas | U | F | Exploring aesthetic characteristics | PR | PQ | CP | Motorists' aesthetic experiences, emotional responses to the parkway's design, landscape variety, scenic beauty, and overall driving experience |
| [29] (22) | Highway | Natural and residential landscapes | U | F | Assessing visual impact | PR | PQ | CTA | Levels of noise emission, two levels of HGV percentage in traffic composition, and three distances to the road |
| [30] (23) | Rural road | A mixture of agriculture and forestry | N | T | Assessing the visual quality | PR | OS | CP | Landscape characters |
| [31] (24) | Rural road | A variety of landscapes, such as agricultural fields, forests, and water bodies | U | F | Assessing the visual quality | PR | OS | CP | Agriculture in the forefront with forest in the background, bare ground in the forefront with forest in the background, and so on |
| [32] (25) | Urban street | A mix of modern and historic elements in an urban setting | U | F | Exploring visual preferences | PR | S | CP | Complexity, coherence, and imageability |
| [33] (26) | Urban street | Poplar planting | N | T | Understanding visual preferences | PR | S and ONI | CP | Naturalness, variety, impressiveness, eye-catchiness, harmony, interest, and excitement |
| [34] (27) | Urban street | A mix of residential, commercial, and administrative areas | NO | T | Assessing visual quality | ER (CM) | - | CTA | Objective indicators and subjective indicators |
| [35] (28) | Urban street | A mix of high-rise buildings, commercial districts, skyscrapers, parks, and so on | NO | T | Analyzing visual perception | PR and ER | S | CP | Salient region saturation (SRS), visual entropy (VE), green view index (GVI), and sky-openness index (SOI) |
| [36] (29) | Urban street | Urban greenery alongside roads | N | T | Assessing visual quality | PR | S | CP | Complexity, interference to coherence, stewardship, naturalness, and beauty impression |

**Table A1.** *Cont.*

| Ref. No. (No) | Type of Road | Landscape Character in Which the Road Is Embedded | Viewers | Views | Purpose of the Study | Assessment Method | Response Format | Media Used to Represent the View | Criteria or Variables |
|---|---|---|---|---|---|---|---|---|---|
| [37] (30) | Urban–rural road | Agricultural land | N | T | Understanding landscape preferences | PR | OS | BWP | Typical development, development with trees, and natural additions like trees and prairie plants |
| [38] (31) | Scenic road | Sightseeing locations, nature and woods, fields, water bodies, and mountains. | N | T | Assessing driving experience | PR and ER | PQ | CP | Sightseeing, nature and woods, fields, water, and mountains |
| [39] (32) | Highway | Urban | NO | F | Assessing landscape character | ER | - | - | Various factors related to green infrastructure |
| [40] (33) | Urban street | Not detail the specific surrounding landscape character | NO | T | Assessing landscape character | ER (CM) | - | CTA | Security, depression, vitality, and aesthetic perceptions |
| [41] (34) | Urban street | None | NO | - | Assessing visual quality | ER | - | - | Identifying and describing twelve visual characteristics |
| [42] (35) | Major roads | Not detail the specific surrounding landscape character | U | F | Assessing landscape character | PR and ER | OS | CTA | Variety, aesthetic of flow, legibility, and orientation |
| [43] (36) | - | - | - | - | - | - | - | - | - |
| [44] (37) | - | - | - | - | Conceptualizing what landscape values mean in practice | ER | - | - | Aesthetic, economic, natural significance, recreation, cultural significance, and intrinsic values |

Notes: Viewers: Users abbreviated as U (e.g., drivers, passengers, pedestrians/bicyclists, etc., with views from the road). Neighbors abbreviated as N (e.g., residents, recreation participants, pedestrians/bicyclists not using the road, students at school, shoppers, etc., with views toward the road). None (without users) abbreviated as NO. Views: From the highway, urban street, rural road, etc., abbreviated as F. Toward the highway, urban street, rural road, etc., abbreviated as T. Assessment method: Public ratings abbreviated as PR. Expert ratings abbreviated as ER. Public ratings and expert rating abbreviated as PR and ER. Computer model (e.g., GIS) abbreviated as CM. Response format: Paper questionnaire abbreviated as PQ. Survey abbreviated as S. Online survey abbreviated as OS. On-site investigation abbreviated as ONI. Slide abbreviated as SL. Media used to represent the view: Color photographs abbreviated as CP. Black and white photographs abbreviated as BWP. Photo-based method abbreviated as PM. Computer technical assistance abbreviated as CTA. Slides abbreviated as SL. Observation abbreviated as OB.

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
