# Peer review of "How Highway Landscape Visual Qualities Are Being Studied: A Systematic Literature Review"

_land, doi:10.3390/land13040431_

Round 1

Reviewer 1 Report (Previous Reviewer 1)

Comments and Suggestions for Authors

General Comments

1.     I suggest that the table in Appendix A be further revised. My suggestion is that there be additional sections. In each section there can be several responses that are coded by a letter or two and  explained in a note after the table. The sections are:

·      Viewers. Users (e.g., drivers, passengers, pedestrians/ bicyclists, etc. with views FROM the road). Neighbors (e.g., residents, recreation participants, pedestrians/bicyclists not using the road, students at school, shoppers, etc. with views TOWARD the road).

·      Views. Are the views FROM or TOWARD the highway?

·      Response format. Paper questionnaire, in-person interview, online survey, etc.

·      Media used to represent the view. On-site, still color photographs, video, animated computer graphic, etc.

·      Variables. Preference/scenic quality, naturalness, openness, diversity, coherence, legibility, mystery, and imageability. I would anticipate other variables: Form, line, color and texture contrast. Vividness, intactness and unity (the FHWA (1981/1990) indicators).

·      Assessment method. Public ratings, expert ratings, computer (e.g., GIS) model.

·      Landscape character in which the road is embedded.

·      Purpose of study. Assess existing views, visual impact, landscape character, etc.

2.     It is still unclear why this review is being conducted—see my first general comment in the first submission review. The response to the review comment is: “The main aim of this article is to develop a framework for having experts and the general public work together to assess highway landscapes employing relevant visual perception variables.” It would seem that a clearer statement would be to identify the perception variables and their effect on preference for highway landscapes. This is why I believe that Appendix table needs to be updated as described above. 

3.     A secondary purpose might be to compare the perception variables in the landscape perception literature to variables used in expert approaches. How many times are they directly compared?

4.     I am missing why studies of the highway landscape is any different from other landscape studies. I think that the difference is found in the linear nature of highways and the movement along them. In particular, movement and sequencing are distinguishing features of the highway experience. High-speed movement on a highway effects viewer attention and other aspects of their visual behavior. Landscape features come into and pass out of view. I would think that this is something to highlight—which studies account for this and how?

5.     The “highway landscape” is still not defined. Is the highway landscape the sum of all areas visible from the highway as well as all the areas with views of the highway (e.g., the area of visible effects)? Are the “viewers” only users of the highway, or do they include people who are not using the highway but see it (e.g., see line 82).Or is it the highway-project’s construction boundary? Could you use “environment” or “area” instead of “landscape”? Maybe it does not really matter.

6.     I think sometimes the terms “valid” and “reliable” are loosely used in this article. Simply making the assertion does not make it so; there need to be research results to support the statement. Valid means that the variables measure what they purport to measure (e.g., is a 5-point rating scale of “coherence” a valid measure of “preference”?) (see line 330). Reliable means that when the method is repeated in the same way by similarly qualified people the results are the same. In general, reliability increases as the number of assessors increases. For methods that involve rating scale, expert judgements are often less reliable because they normally use only one or two assessors; public methods are more reliable because they often use large groups (e.g., more than 50 people) and the variables are more salient (e.g., preference or scenic quality compared to coherence or mystery).

7.     I think that the literature review is a different article than the framework in Figure 2 (line 462). In particular, I do not see how this framework is the synthesis of the literature. For instance, there are landscape attributes in the literature that do not seem to be covered in this framework. What is the support for believing that the hybrid approach will be more reliable than either the public or expert approaches? It seems more like a proposal based on the author’s judgement of what is important.

8.     If the framework is the objective of this article, then the frameworks mentioned in line 458 should be the focus of the review (i.e., not so much the results of particular indicators, but which ones are most reliable and valid for the framework’s proposed purpose). There should be an explanation of their strengths and weaknesses, and why you believe that. Then how the framework in Figure 2 addresses that, for instance how are the public and expert approaches integrated? Ideally it would be tested.

9.     I whole heartedly agree that “investigating visual preferences encompasses not only aesthetics but also the interaction between sensory experience and environmental perception” (line 494). In some ways the function of a highway—to get from point A to point B—determines the landscape through which it passes. However, even with this limitation there are many design options within the highway project corridor. In particular it is possible to create highways that a joy to drive, or that put the driver to sleep. For instance see: Myers, M.E., 2004. The line of grace: Principles of road aesthetics in the design of the Blue Ridge Parkway. Landscape Journal23(2), pp.121-140.

/\/\/\/\/\/

The following are thoughts concern me, but I do not think there is much to do with them at this point.

10.  I wonder about the criteria for identifying the literature. Some of the studies do not include landscape preferences (e.g., [27] Chamberlain & Meitner is a type of GIS visibility analysis), while some studies that investigate highway landscape preferences are missed (e.g., Myers, M.E., 2006. Power of the picturesque: Motorists’ perceptions of the Blue Ridge Parkway. Landscape Journal25(1): 38-53. Perhaps it is too old?) Another article that is missed surprises me-- Bishop, I., 2013. Analysis of sequential landscape experience. Peer-Reviewed Proceedings of Digital Landscape Architecture 2013, pp.55-64. I particularly miss some older seminal studies, such as: 

·      Appleyard, D., Lynch, K., & Myer, J. R. 1964. View from the Road. 

·      Craik, K.H., 1975. Individual variations in landscape description. In Zube, Brush & Fabos (eds.) Landscape Assessment: Values, Perceptions and Resources. Stroudsburg (Pennsylvania), Dowden, Hutchinson and Ross.

Generally speaking, the professional interest in highway aesthetics has to do with design, not assessment methods, per se. As practicing professionals, their ideas are normally not presented in academic journals, but in books. For instance: 

·      McCluskey, J., 1979. Road form and townscape. Publication of: Architectural Press Limited

·      Halprin, L. 1966. Freeways. Reinhold.

11.  Is there any reason to believe that most of the reviewed articles will have any real-world effect on highway aesthetics, either generally or for specific highways, or are they simply academic studies? Line 284 states that “Understanding these influences is essential for optimizing landscape management to cater to diverse preferences and enhance highway travel's visual and experiential quality.” Do you really believe that? Are there any examples where this has occurred?

It may be that it is simply a review of the academic literature on viewer’s preferences, but then I would expect it to be limited to landscape perception studies. If the purpose was to investigate how highways change the landscape (i.e., visual and landscape impact assessment), then I would expect to see visual impact assessment procedures designed specifically for roads, such as:

·      U.S. Department of Transportation, Federal Highway Administration (FHWA). 1981/1990. Visual Impact Assessment for Highway Projects. (Publication No. HEV-22/3-81 (650), FHWA-RE-90-007).

·      Caltrans. 2023. Visual impact assessment handbook. California Department of Transportation. https://dot.ca.gov/-/media/dot-media/programs/design/documents/via-handbook--a11y.pdf (Accessed on February 26, 2024)

·      Clamp, P. E. (1976). Department of the Environment. Evaluation of the impact of roads on the visual amenity of rural areas. (Research report 7). London, UK: Department of the Environment.

Other purposes might be to inform the designation of scenic highways, or to guide highway construction mitigation. 

Understand that some of the above references may be too old to interest you, but there are updated versions (which may or may not be improvements).

Specific Comments (by line number)

There are no major language issues, but many small text edits would help this paper. I have not attempted to address those and suggest engaging a native-English speaking editor.

002   Perhaps the title should be revised to: How are Highway Landscape Visual Qualities being Studied”? The reviewed articles are not all about “viewers’ preference.” In particular some are about expert assessments where the evaluation is not about preference, but factors that are described as “objective.” At least one article is about visibility.

033   Change “Highway” to “Highways”.

052   Ernawati [9] does not appear to discuss “emotional reactions”. He it looking at the Kaplans’ cognitive variables.

220   Is “deep learning” just a type of “machine learning technique”?

232   What is “d F” and in the next line, “L”?

278   What does “of 11.6  has” mean?

279   Views at 300 meters is “slightly intensified” compared to what?

301   Landscape perception studies in general and [4] in particular use mean ratings; it is not a “consensus” procedure.

313   Does the study actually test reliability and accuracy?

321   Why is the expert approach “fundamentally an objective methodology”?

336   Depending on the approach, deep-learning should be reliable (gives the same result if the conditions are the same), but it may not be valid.

359   What “standard”?

409   Landscape character assessment [41] has a strong cultural interpretation component as well as these more physical aspects.

428   If it is true that “that, in general, the more open the view, the higher the preference,” then vast plains without trees or structures would be the most preferred landscape. I suspect that the relation of openness to preferences is an inverted-U, and there is probably an interaction with the speed of travel.

441   “Diversity, coherence, legibility, mystery” are cognitive attributes (Kaplan & Kaplan.1982. Cognition and Environment. Praeger.) Why is “diversity” included and “complexity” missing from this list? The Kaplan variables tend not to be used in real-life projects because, among other reasons, they are difficult to judge reliably ([8] Stamps 2004). Examples of “highway-landscape visual composition” attributes are vividness, intactness, unity  and the contrast elements form, line, texture & color.

457   Provide citations for these frameworks.

470   Why is the framework more reliable?

533   Replace “reliable” with “valid”.

Comments on the Quality of English Language

The English is acceptable.

Author Response

Dear reviewer:

Reviewer 2 Report (Previous Reviewer 2)

Comments and Suggestions for Authors

The revised paper has significantly improved but changes are still needed before it can be published.

Major comments

Methods section needs a subsection on analysis, specifically how you derived the themes for your results subsections. When I read through your results it’s unclear to me how the different subsections differ from one another, particularly subsections 3.1, 3.2, and 3.3. If you describe your process for thematic development in Sec 2, at the beginning of Sec 3 you should then briefly describe the main themes for each of the subsections that follow. By being clear on this, the titles of these subsections might also be revised so they are more clearly distinct from each other.

The text within the subsections sometimes seems duplicative. For example in 3.2 you discuss the interactions between traffic noise and highway visual perception (LL208-9), and in 3.3 you discuss traffic noise and visual preferences again (LL276-278). If the purpose of the different subsections is made clearer there may be a justification for talking about noise-visual preference interactions but as it now reads it seems poorly organized and redundant.

With respect to this last point, Table 6 is not referenced in the text and its title is vague, so I’m not sure how it related to your analysis or to the subsection of the results where it’s located.

Editorial comments

L38 solid natural ecology is ambiguous

LL43-44 Sentence logic seems circular

LL60-61 “In recent years, the preferences and perceptions of landscapes have changed significantly due to rapid urbanization [11].” I somewhat disagree with this statement and while [11] may present data to support it other work generally shows landscape perceptions and preferences remain stable over time. On the other hand, these perceptions and preferences are no doubt impacted by urbanization and other forms of landscape change so I think it would be better to rephrase the sentence as ““In recent years, people’s preferences and perceptions of landscapes have been impacted significantly due to rapid urbanization.”

LL205 Revise “The exploration of visual quality extends beyond aesthetics” to read The exploration of landscape preference extends beyond visual quality

LL225 Revise “Overall, these studies affirm that all respondents, viewed as viewers of the roadside…” to “Overall, these studies affirm that all respondents, considered as viewers of the roadside…”

LL231-2 Something is wrong with this sentence: “This approach allows for a more detailed and integrated understanding of highway landscape percept d F with vegetation were better than Group L, where artificial landscapes dominate.”

LL236 Table 6 is not referenced in the text and its title is vague, so I’m not sure how it related to your analysis.

Sec 3.4 title is wordy Assessment Approaches of Road Landscape Visual Preference as a Model for Highway Landscape Visual Preference

Subsections 3.5 and 3.6 have the same title:

3.5. Assessment Variables of Road Landscape as a Model for Highway Landscape Visual Preference

3.6. Assessment Variables of Road Landscape as a Model for Highway Landscape Visual Preference

Fig 2 In your text you describe and advocate for a hybrid approach to highway landscape assessment but your figure calls it an integration approach—this should be changed to hybrid for clarity and consistency.

Comments on the Quality of English Language

Wording can be improved overall

There appears to be considerable redundancy in text

Author Response

Dear reviewer:

Reviewer 3 Report (New Reviewer)

Comments and Suggestions for Authors

The authors define a clear objective for the paper, which is to review the literature, and they do so. They also recognize the limitations of their study. The paper offers a useful cataloguing of literature from which a framework is drawn. It is a useful reference for its purpose. It is by no means a revolutionary study and I suspect its audience will be limited, but it nevertheless does highlight a role played by highways that is not typically discussed in the literature.

Comments on the Quality of English Language

Only minor proofreading required.

Author Response

Dear reviewer:

Reviewer 4 Report (New Reviewer)

Comments and Suggestions for Authors

This is an interesting paper, presenting an innovative approach to the analysis of landscape preference in cases of landscapes involving highways. .

The number of papers examined is enough for the purpose of this reasearch, the paper itself is well written and, given its relevance to the scope of the journal, it is suitable and ready for publication. However, there are some margins for expaning a bit more (i.e. in the Discussion of the results) on two keywords that have been referred to repeatedly in the papers that were examined : landscape complexity and visual entropy. The authors may take into consideration that, in the case of the papers they examined, it is both what is described (and analyzed in depth with respect to visual and aesthetic aspects) as “spatial entropy" and "spatial complexity". See Papadimitriou, F. (2020): Spatial Complexity, Cham: Springer and Papadimitriou, F. (2022): Spatial Entropy and Landscape Analysis, Wiesbaden: Springer.

Author Response

Dear reviewer:

Round 2

Reviewer 1 Report (Previous Reviewer 1)

Comments and Suggestions for Authors

Summary

I think the peer-review process has reached the point where I have little left to offer the author of this article. I appreciate the author’s efforts to respond to past reviews, in particular the more complete profile of the literature in the appendix. I hope that the author feels the per-review process has in proved the article.

General Comments

1.     I anticipate that Land will print the appendix on horizontal pages to make it more readable. I suggest that reading through it to catch simple copy editing errors, such as consistent use of capital letters.

Specific Comments (by line number)

There are no major language issues, but many small text edits would help this paper. I have not attempted to address those and suggest engaging a native-English speaking editor.

002   I accept that it is the author’s choice to choose an appropriate title. However, I still suggest the title should be revised to: “How highway landscape visual qualities are being studied.” The reviewed articles are not all about “viewers’ preference.” In particular some are about expert assessments where the evaluation is not about preference, but factors that are described as “objective.” At least one article is about visibility.

316   Change “on a specific scale” to on a scale of “1 to 100”.

463   While it may be true that “the study's results [10] have shown that naturalness does not significantly improve the ability to predict preferences,” I hope that the author recognizes that this is an unusual finding (e.g., Kaplan, R. and Kaplan, S. 1989. The experience of nature: A psychological perspective. Cambridge Univ. Press.).

Comments on the Quality of English Language

If possible, it might help to have a native English speaker review it one more time. 

Author Response

Dear reviewer:

Reviewer 2 Report (Previous Reviewer 2)

Comments and Suggestions for Authors

Accept

Author Response

Dear reviewer:

Thank you very much, for the advice you gave earlier. It is very helpful for me.

This manuscript is a resubmission of an earlier submission. The following is a list of the peer review reports and author responses from that submission.

Round 1

Reviewer 1 Report

Comments and Suggestions for Authors

Summary

This paper did an article search of three standard databases to find the published literature about preference and the highway landscape. 246 articles were identified and further screened by three criteria: (1) publication between 2013 and 2023, (2) publication in in English in a scientific journal (i.e., per-reviewed), and (3) identified as a research paper. 22 papers met these criteria and a review of their references resulted in the addition of four more papers.

While the paper discusses these studies, it is difficult to identify a systematic method used to conduct the review and the comprehensive results of applying that method.

General Comments

1.     It is unclear why this review is being conducted. On the one hand there is general expectation that this literature contributes somehow to the planning, design and maintenance of highways. However, there is no particular reason to believe this to be true—e.g., what evidence is there that these peer-reviewed journal publications influenced a highway landscape? On the contrary, this activity has long been documented in the so called “gray-literature” as scenic assessment methods [1], design guidelines [2], and visual impact analysis methods [3, 4]. Most implemented highway assessments use expert evaluation; it would be very rare for a government agency to use landscape perception methods. For example, this special issue is associated with the 2023 Visual Resource Stewardship conference, and there were a number of agency presentations about road and highway projects that will never find their way into a peer-reviewed journal. The point of this comment is to request that the purpose of the review be clearly stated.

2.     Landscape assessment and landscape perception have been recognized as topics of research interest since the late-1960s [5]. An important question for this review should be to identify how the study of highways differs from the standard landscape assessment/perception study. What comes immediately to mind is that landscape perception on highways is a dynamic sequential experience. How is that incorporated into the study. For instance, is the scenic quality equally important at the beginning, middle and end of a trip? Is it important to have visual diversity, or should all views be based on the same guidelines (e.g., as open as possible) even if that becomes monotonous? 

3.     How landscape vis-à-vis the highway (or other roads) needs to be clearly defined. Landscape is not defined until line 215, where the European Landscape Convention definition seems to be accepted. If this is relevant, it needs to be in the Introduction, not the Results section. In identifying landscape character areas, it would be highly unusual that to separate the highway from the landscape in which it is located [6]. The paper slips between using the terms landscape and environment synonymously, which they are not. It is suggested that the immediate area that is part of the highway be referred to as the “highway environment” and that the surrounding visible area be referred to as “surrounding landscape” or “visible landscape.”

4.     There is no attempt to define “users,” perhaps better called “viewers.” There are two distinctly different groups: there are viewers traveling on the highway looking out into the surrounding landscape, and there are users with views of the highway who are not on it. Then within these two general groups there are a number of different users. People on the highway may be drivers or passengers; commuters in personal vehicles or public transportation; they may be professional drivers or tourists; they may be on a bicycle or walking. Those with a view of the highway may be residents in a high-rise apartment building, or hikers in a national park, or children playing in a neighborhood. The affect and importance of landscape visual quality is different for different types of users.

5.     The paper’s objectives as stated in line 75 to 77 appear to be “to systematically review the existing literature on the evaluation of preference for highway landscapes and subsequently formulate comprehensive criteria for such evaluations.” What I expect from table A1 is a list of the 26 selected articles identified only by their citation number. Then the table should summarize (1) the type of roadway studied, (2) the surrounding landscape character, (3) the types of users considered, (3) the purpose of the study: e.g., inventory, design guidelines, visual impact assessment (4) whether the assessment surveyed user perceptions, or was an expert assessment, (5) the criteria or variables used in the assessment, (6) the most important results. This might result in a lot of text, so a scheme of abbreviations may need to be developed. For instance, in the criteria/variable section it might be: D = diversity, C = coherence, and L = legibility. Then the results discuss each of these major review topics (the final topics may be different from those suggested here) in more detail than presented in the table. All of these terms need to be defined, or references given for how they are defined. It may be important to recognize that some terms have different definitions for different researchers (e.g., Lynch and the Kaplans define legibility differently).

6.     A problem for this this review is that there are significant number of unique conditions when one considers the diversity of landscapes, types of roads, types of users, assessment criteria or variables. There are simply too few articles to make confident generalizations beyond those already identified from decades of landscape perception research. For instance, [2] studies a highway in Tehran city—is it reasonable to apply the results to rural landscapes? However, this shortcoming can recognize in the paper, which can only report what has been learned and suggest what needs to be done in future research to flesh out these findings. Given this problem, is it reasonable to propose a unified comprehensive procedure and comprehensive criteria? Do not overstate the findings.

7.     Finally, there appears to be a major problem with accurately attributing findings to the cited references. If this paper is resubmitted, it is necessary to explain why this happened. 

Specific Comments (by line number)

There are no major language issues, but many small text edits would help this paper. I have not attempted to address those and suggest engaging a native-English speaking editor.

54     Reference [9] does not appear to have anything to do with landscape perception or preferences.

57     Reference [11] does not appear to consider how mass transportation leads to the construction and expansion of roads and highways.

160   Reference [6] has some discussion of the literature as it relates to “attention [being] more consistently directed towards distant views,” but it is not part of their findings.

223   Public perceptions may be subjective, but they can be investigated using relatively objective methods and both the reliability and validity of the results can be determined. This is also just as true for most expert approaches, which are typically based on judgements using rating scales or less rigorous descriptions.

234   Reference [29] does not appear to have a finding about whether participants understood the questions correctly. This study used a head mounted display and some people complained of dizziness and their participation was therefore terminated [29, section 3.2].

253   Reference [19] is a literature review from diverse sources; not an empirical study. There does not appear to be a general finding that the expert approach “is flawed because it relies solely on expert knowledge and is based on underdeveloped criteria and invalid assessment parameters.” The same can be said for [31] which finds that four computer-determined indices can effectively reflect the visual attributes of streets.

260   The paper states that “Researchers have attempted to combine both methods,” by which the public perception and expert approaches from lines 223 to 259 are assumed to be the methods being referenced. This is reinforced by the text at line 221. Reference [26] uses “mixed methods,” but the meaning is to use both quantitative and qualitative methods, not public perception and expert approaches, which can be either quantitative or qualitative.

308   Reference [24] does not seem to support the generalization in this paper that “the narrow visual space created by too much road vegetation is usually less preferred by people.” To the contrary, [24] finds that “notably, in the negative group, the preference for the view with enclosed horizons was higher than that with partially open horizons.” This is the only use of “narrow” in article [24].

References (examples of many possible)

[1]    New York State, Department of Environmental Conservation. (n.d.) Preserving New York State scenic roads.

[2]    US National Park Service. (1990). Road character guidelines: Sequoia & Kings Canyon National Parks.

[3]    Clamp, P. E. (1976). Department of the Environment. Evaluation of the impact of roads on the visual amenity of rural areas. (Research report 7). London, UK: Department of the Environment.

[4]    U.S. Department of Transportation, Federal Highway Administration (FHWA). 1981/1990. Visual Impact Assessment for Highway Projects. (Publication No. HEV-22/3-81 (650), FHWA-RE-90-007).

[5]    Gobster, P.H., Ribe, R.G. and Palmer, J.F. (2019). Themes and trends in visual assessment research: Introduction to the Landscape and Urban Planning special collection on the visual assessment of landscapes. Landscape and Urban Planning, 191, p.103635.

[6]    Tudor, C. (2014). An approach to landscape character assessment. Natural England.

Comments on the Quality of English Language

While the language is quite understandable, attention needs to be given to the use of articles (i.e., a, the) and number (i.e., singular, plural).

Author Response

Dear reviewer,

Please, check the attached file.

Reviewer 2 Report

Comments and Suggestions for Authors

see file

Comments on the Quality of English Language

typos, formatting issues

Author Response

Dear reviewer,

please, check the attached file. i have resubmitted.
